# Intrauterine growth and the tangential expansion of the human cerebral cortex in times of food scarcity and abundance

Daniel E. Vosberg[1,2,3], Igor Jurisica [4,5,6], Zdenka Pausova [3,7,8] & Tomáš Paus [1,2,8,9,10] ✉

Tangential growth of the human cerebral cortex is driven by cell proliferation during the first and second trimester of pregnancy. Fetal growth peaks in mid-gestation. Here, we explore how genes associated with fetal growth relate to cortical growth. We find that both maternal and fetal genetic variants associated with higher birthweight predict larger cortical surface area. The relative dominance of the maternal *vs*. fetal variants in these associations show striking variations across birth years (1943 to 1966). The birth-year patterns vary as a function of the epigenetic status near genes differentially methylated in individuals exposed (or not) to famine during the Dutch Winter of 1944/1945. Thus, it appears that the two sets of molecular processes contribute to early cortical development to a different degree in times of food scarcity or its abundance.

The growth of the human cerebral cortex begins with the emergence of the ventricular (5th the post-conception week) and subventricular (7th post-conception week) zones[1]. The radial unit hypothesis provides a framework for global and regional expansion of the primate cerebral cortex emanating from these two proliferative zones[2]. In primates, the phase of *symmetric division of progenitor cells* in the proliferative zones during the first trimester is particularly important for the tangential growth through the additions of ontogenetic columns[2]. Ionizing radiation of the (monkey) fetus during early gestation reduces surface area (SA) of the cerebral cortex[3]. The subsequent phase of *asymmetric division* continues to increase the number of ontogenetic columns (and thus SA) later in pregnancy. The growth of SA (but not neurogenesis) continues postnatally, being mostly completed by the end of infancy and remaining stable

thereafter[4,5]. Thus, cortical SA in adulthood provides a snapshot of early developmental processes. Individual variability in cortical SA is genetically and phenotypically related to mental illness, cognitive abilities, and educational attainment[6,7].

Fetal growth accelerates during the second trimester, with the peak values of head circumference, femur length, and abdominal circumference observed in the 16th week of gestation[8]. At birth, the weight of a newborn varies as a function of the pregnancy duration; for infants born at term (37–40 weeks of gestation), birthweights show a normal distribution[9]. Even in individuals born at term, low birthweight predicts mental illness later in life[10]. Birthweight is a function of both the fetus and the mother[11]. The genetics of birthweight can be partitioned into two (statistically) independent components: (1) fetal variants associated directly with the fetal

[1]Centre Hospitalier Universitaire Sainte-Justine, University of Montreal, Montreal, Quebec, Canada. [2]Department of Neuroscience, Faculty of Medicine, University of Montreal, Montreal, Quebec, Canada. [3]Research Institute of the Hospital for Sick Children, Toronto, ON, Canada. [4]Osteoarthritis Research Program, Division of Orthopedic Surgery, Schroeder Arthritis Institute, and Data Science Discovery Centre for Chronic Diseases, Krembil Research Institute, University Health Network, Toronto, ON, Canada. [5]Departments of Medical Biophysics and Computer Science, and the Faculty of Dentistry, University of Toronto, Toronto, ON, Canada. [6]Institute of Neuroimmunology, Slovak Academy of Sciences, Bratislava, Slovakia. [7]Departments of Physiology and Nutritional Sciences, University of Toronto, Toronto, ON, Canada. [8]ECOGENE-21, Chicoutimi, Quebec, Canada. [9]Departments of Psychology and Psychiatry, University of Toronto, Toronto, ON, Canada. [10]Department of Psychiatry, Faculty of Medicine, University of Montreal, Montreal, Quebec, Canada. ✉e-mail: tpausresearch@gmail.com

birthweight; and (2) maternal variants influencing the intrauterine environment and, in turn, the growth of the fetus[12]. Both genetic components appear to correlate with the genetic landscape associated with head circumference, but only the maternal genetic variants correlate with the genetic landscapes associated with cognitive abilities and years of education[12].

## Results

### Examining phenotypic relations between birthweight and cortical surface area

Among 17,263 UK Biobank[13] participants of European ancestry with magnetic resonance imaging (MRI) and birthweight data, we fit multiple regression models, with SA (global or regional [34 regions]) as the outcome variable, birthweight as the explanatory variable, and age at MRI, sex, and the first 10 principal components of genetic ancestry as covariates. The SA values were derived using FreeSurfer and the Desikan-Killiany atlas[14] and summed across the two hemispheres. We found that greater birthweight was associated with larger SA globally and across all 34 regions (Fig. 1). These results are consistent with those reported previously[15–17]. Moreover, we repeated these analyses in a sex-specific manner, identifying positive associations between birthweight and SA across all regions in males (n = 7274) and females (n = 9989; Fig. 1). The effect sizes (β) were significantly higher among females (M[β] = 0.11, SD[β] = 0.034), than males (M[β] = 0.09, SD[β] = 0.027), as assessed with a paired samples t-test (t[34] = 6.57, p = 1.59 × 10⁻⁷).

### Testing associations between polygenic scores for birthweight and cortical surface area

Among 29,047 UK Biobank participants with MRI and genetic data, we assessed associations between cortical surface area and the polygenic scores (PGSs) for birthweight (n = 298,142), and its fetal and maternal components (n = 406,063)[12]. We fit multiple regression models, with birthweight, fetal and maternal PGSs, restricted to genome-wide significant single nucleotide polymorphisms (SNPs) as independent variables, global and regional cortical SA as dependent variables, and adjusting for sex, age at MRI, and the first 10 principal components of genetic ancestry. The PGSs were computed at the genome-wide significant threshold using PRSice-2[18]. As shown in Fig. 2, we found positive associations between SA and birthweight PGS (p_FDR <0.05 for 31/34 regions and globally), as well as between SA and the two genetic components, fetal PGS (p_FDR <0.05 for 25/34 regions and globally) and maternal PGS (p_FDR <0.05 for 28/34 regions and globally). There were a greater number of statistically significant findings among males (maternal PGS: p_FDR < 0.05 for 29/34 regions and globally; fetal PGS: p_FDR < 0.05 for 22/34 regions and globally; n = 14,142) than females (maternal PGS: p_FDR < 0.05 for 1/34 regions; fetal PGS: p_FDR < 0.05 for 14/34 regions and globally; n = 14,905; Fig. S1). These sex-specific differences were supported by statistically significant differences in the effect sizes across regional and global SA for both the maternal (t[34] = 9.66, p = 2.80 × 10⁻¹¹) and fetal (t[34] = 2.44, p = 0.02) PGSs. Additionally, given the known relationship between height and head circumference[19], we have examined the correlations between height

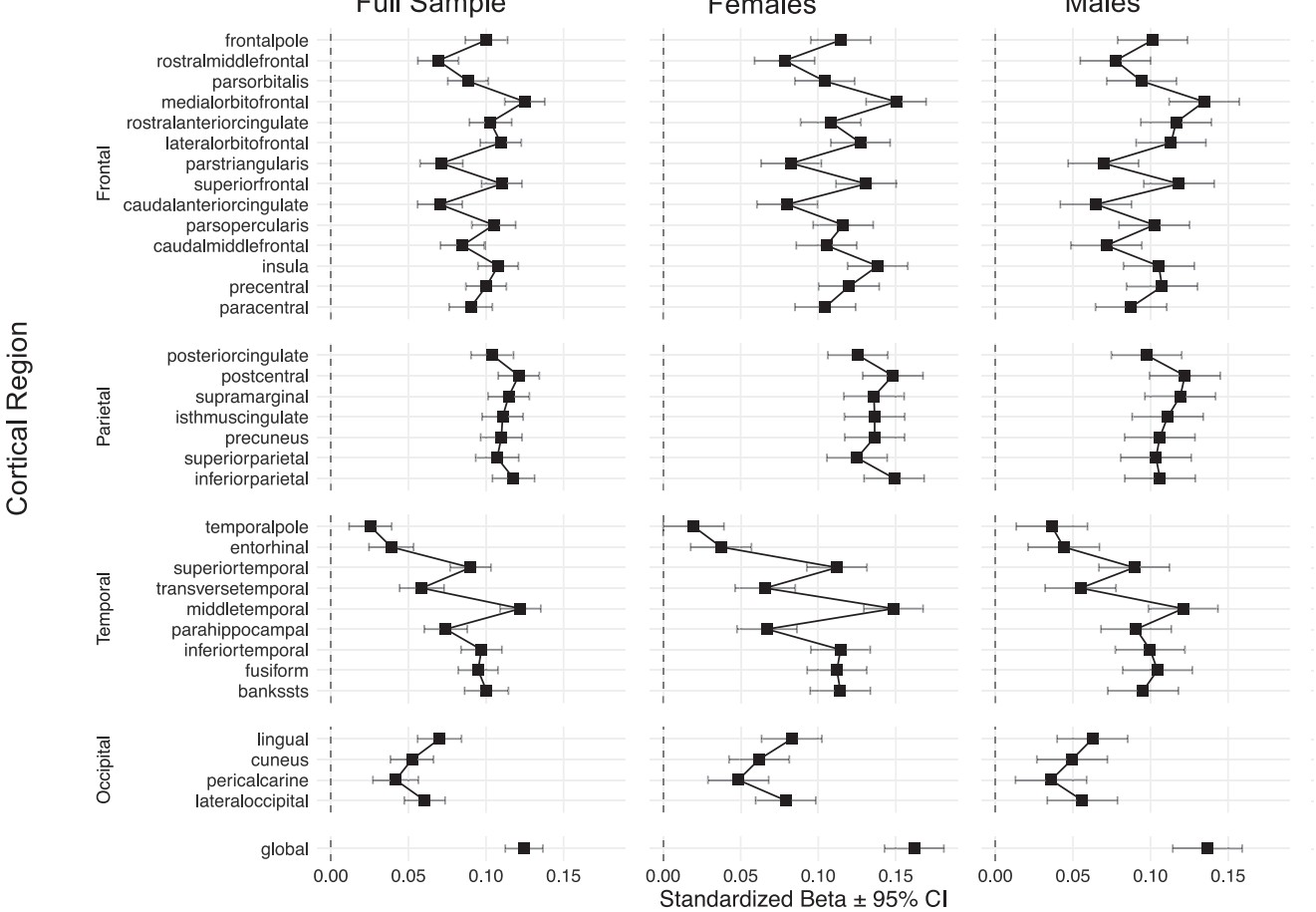

**Fig. 1 | Associations between birthweight and cortical surface area.** Two-sided multiple linear regression models were fit, assessing the associations between birthweight and regional (and global) cortical surface area in the full sample (n = 17,263), males (n = 7274), and females (n = 9989), and standardized effect sizes (β) and 95% confidence intervals are shown. All associations were positive and significant following an adjustment for the False Discovery Rate (FDR) for the 34 regions and globally (filled squares).

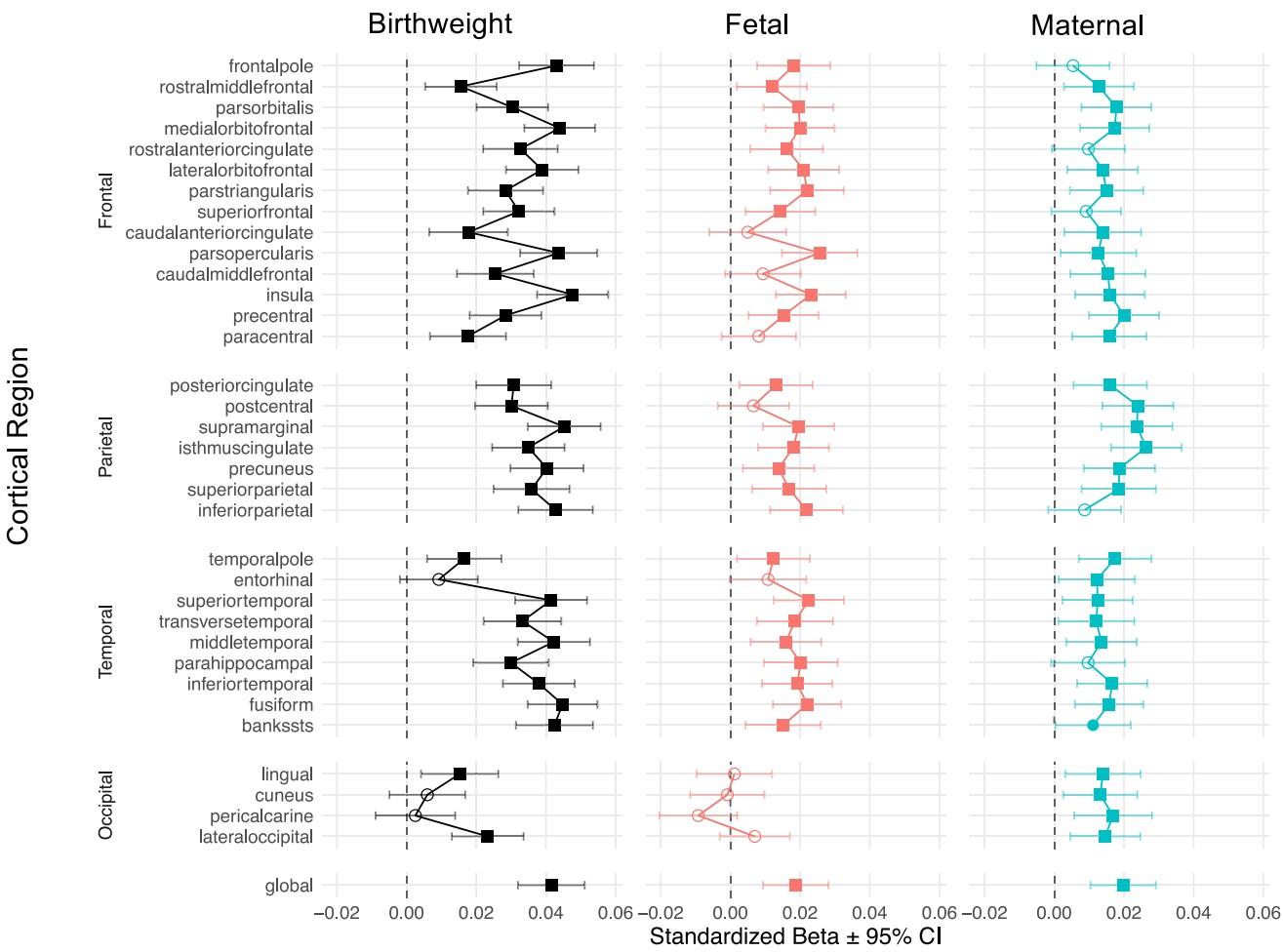

**Fig. 2 | Associations between polygenic scores (PGSs) for birthweight, fetal, and maternal variants, and cortical surface area.** Two-sided multiple linear regressions were conducted, between PGSs for birthweight (black), fetal (red) and maternal (cyan) variants and cortical surface area ($n = 29{,}047$); standardized effect sizes ($\beta$) and 95% confidence intervals are shown. The significance levels are indicated by filled squares ($p_{FDR} < 0.05$), filled circles ($p < 0.05$), and open circles ($p > 0.05$).

PGS and birthweight PGSs (fetal and maternal). We have also re-analyzed the correlations between the cortical surface area and birthweights PGSs after adjusting the latter for the height PGS. These results are provided in the Supplementary Material (Fig. S2). The main findings remained the same after this adjustment.

Considering the secular trends in growth[20], we partitioned our sample by year of birth to investigate potential birth-year effects. Given the distribution of our sample by year of birth, with the majority born between 1943 and 1966, we excluded participants outside of this birth range and, to ensure sufficient power, generated four birth-year periods each comprising a span of 6 years (i.e., 1943–1948 [$n = 6888$], 1949–1954 [$n = 8048$], 1955–1960 [$n = 6349$] and 1961–1966 [$n = 4971$]; Fig. S3). As shown in Fig. 3, in the sex-combined sample, we observed associations between maternal (but not fetal) birthweight PGSs and SA for the 1943–1948 period ($p_{FDR}$ and/or nominal <0.05 for 17/34 regions and globally) and, albeit less strongly, for the 1961–1966 period ($p_{FDR}$ and/or nominal $p < 0.05$ in 15/34 regions and globally). Associations between fetal birthweight PGS and SA were (largely) not statistically significant in these two periods of birth. Moreover, we identified associations between fetal birthweight PGSs and SA for the 1949–1954 period ($p_{FDR}$ and/or nominal <0.05 for 17/34 regions and globally) and the 1955–1960 period ($p_{FDR}$ and/or nominal < 0.05 for 18/34 regions and globally). Associations between maternal birthweight PGS and SA were (largely) not statistically significant in these two periods of birth.

The effects observed in the 1943–1948 period appeared driven by males (Fig. S4A) who demonstrated associations between maternal

PGSs and SA, with greater effect sizes among males compared with females ($t_{[34]} = 5.70$, $p = 2.10 \times 10^{-6}$); these associations were not statistically significant among females ($n = 3117$; Fig. S4B). Finally, we evaluated birth-year period $x$ PGS interactions vis-à-vis SA. There were nominally significant interactions in (mostly) frontal cortical regions with the maternal but not the fetal PGS, especially in males, demonstrating that relative to the 1943–1948 birth period, the cohort of individuals born 1955–1960 had weaker associations between the maternal PGS and SA (Fig. S5A). No such interactions were observed in the sex-combined cohort for the fetal PGS and SA (all $p \geq 0.06$), while a few nominally significant interactions were observed in each sex individually (Fig. S5B).

## Constructing and testing polygenic scores for epigenetic response to famine

Given the above birth-year differences in the associations between cortical surface area and, respectively, maternal PGS (1943–1948 and 1961–1966) and fetal PGS (1949–1954 and 1955–1960), we sought to explore the possibility that these are related to the individual's exposure to food restriction during their gestation or infancy (1943–1948). We also reasoned that, given the similarity of these associations between the 1943–1948 and 1961–1966 cohorts, part of this observation might be related to a transgenerational transmission of this exposure to offspring born to parents who experienced food scarcity during WWII (1939–1945) during their gestation. Among the 1961–1966 cohort, their mothers were born between 1917 and 1950 (1st quartile:

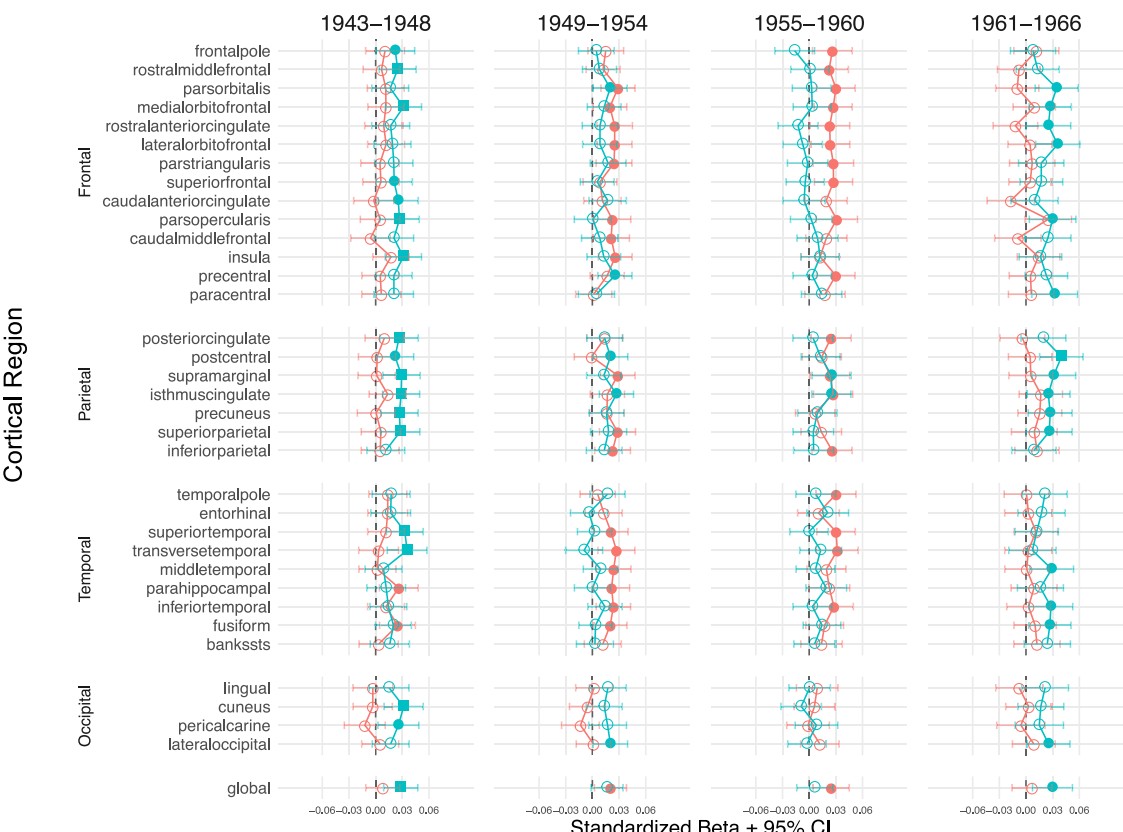

**Fig. 3 | Associations between polygenic scores (PGSs) for fetal and maternal variants and cortical surface area, stratified by year of birth period.** Two-sided multiple linear regressions were conducted, between PGSs for fetal (red) and maternal (cyan) variants and cortical surface area, stratified by year of birth period, and standardized effect sizes (β) and 95% confidence intervals are depicted. The analyses were conducted in the following birth year periods: 1943–1948 ($n = 6888$), 1949–1954 ($n = 8048$), 1955–1960 ($n = 6349$), and 1961–1966 ($n = 4971$). The significance levels are indicated by filled squares ($p_{FDR} < 0.05$), filled circles ($p < 0.05$), and open circles ($p > 0.05$).

1933, median: 1937, 3rd quartile: 1940). Moreover, among the mothers of the 1961–1966 cohort, 34% were born during WWII, while 13% were born between 1943 and 1948 (Fig. S6). As described in detail in the 1946 report by the UK Ministry of Food[21] the UK population experienced both quantitative (e.g., lower energy intake) and qualitative (e.g., shift to bread and potatoes) change in food availability during and shortly after World War II (WWII). Food rationing began in 1940 and lasted until ~1948; note, however, that pregnant women and infants received additional rations. In Nordic countries, comparable changes in food consumption during WWII (vs. pre-war period) were associated with lower height and weight of Norwegian and Finish school-age children (7 to 13 years of age) in the war vs. pre-war period[22]. The period of severe food restrictions experienced by the population of Western Netherlands during the Dutch Winter of 1944/1945 represents a well-studied case of both direct and transgenerational effects of this adversity on the health of prenatally exposed individuals (e.g.[23,24,]) and their offspring[25,26], possibly mediated by a differential methylation of their DNA[27]. For this reason, we have constructed a polygenic score that predicts methylation status of cytosines in CpG dinucleotides known to be differentially methylated in the individuals exposed (vs. not)–during gestation–to famine during the Dutch Winter[28] (Fig. S7A). As such, this PGS represents an instrumental variable[29] that models the effect of famine on DNA methylation; by design (Fig. S7B), individuals with high (vs. low) PGSs can be considered to be (virtually) famine-exposed (vs. non-exposed). Henceforth, we term this polygenic score the "famine PGS". Given the observed birth-year differences in the associations between cortical surface area and the maternal and fetal PGSs (Fig. 3), we predicted that individuals born at the time of food scarcity (1943–1948 cohort) and the "offspring" of individuals born

between the first and second world wars (1961–1966 cohort) would show different effects of the famine PGS (on the maternal/fetal PGS–SA relationships) than those born when food was abundant (1949–1954 and 1955–1960 cohorts). As shown in Fig. 4 (details in Fig. S8 [all], S9 [males] and S10 [females]), this was the case. The two cohorts in which the maternal (vs. fetal) variants dominated the correlations with SA (1943–1948 and 1961–1966) differed from the other two cohorts in which fetal (rather than maternal) variants dominated these correlations (1949–1954 and 1955–1960). Furthermore, the same pattern of differences (in the PGS-SA correlations) between the "high" and "low" famine PGS subgroups was observed within each of the distinct cohort pairs (exposed and non-exposed). Thus, for the relationship between maternal genetic variants and cortical surface area (Fig. 4, top row), the effect of these variants was lower in individuals with high (vs. low) famine PGS in the (exposed) 1943–1948 and 1961–1966 cohorts while it was higher (in the same comparison) in the (non-exposed) 1949–1954 and 1955–1960 cohorts. On the other hand, for the relationship between fetal genetic variants and cortical surface area (Fig. 4, bottom row), the effect of these variants was higher in individuals with low (vs. high) famine PGS in the (non-exposed) 1943–1948 and 1961–1966 cohorts while it was lower (in the same comparison) in the (exposed) 1949–1954 and 1955–1960 cohorts. Thus, our analysis suggests that in the cohorts born when food was abundant, the relationships between both the maternal and fetal PGS and cortical surface area are enhanced in individuals with high (vs. low) famine PGS. In the cohorts exposed to food scarcity (either directly or trans-generationally), the opposite is true: the combination of this adversity with high famine PGS appears to diminish the relationship between maternal and fetal PGSs and cortical growth.

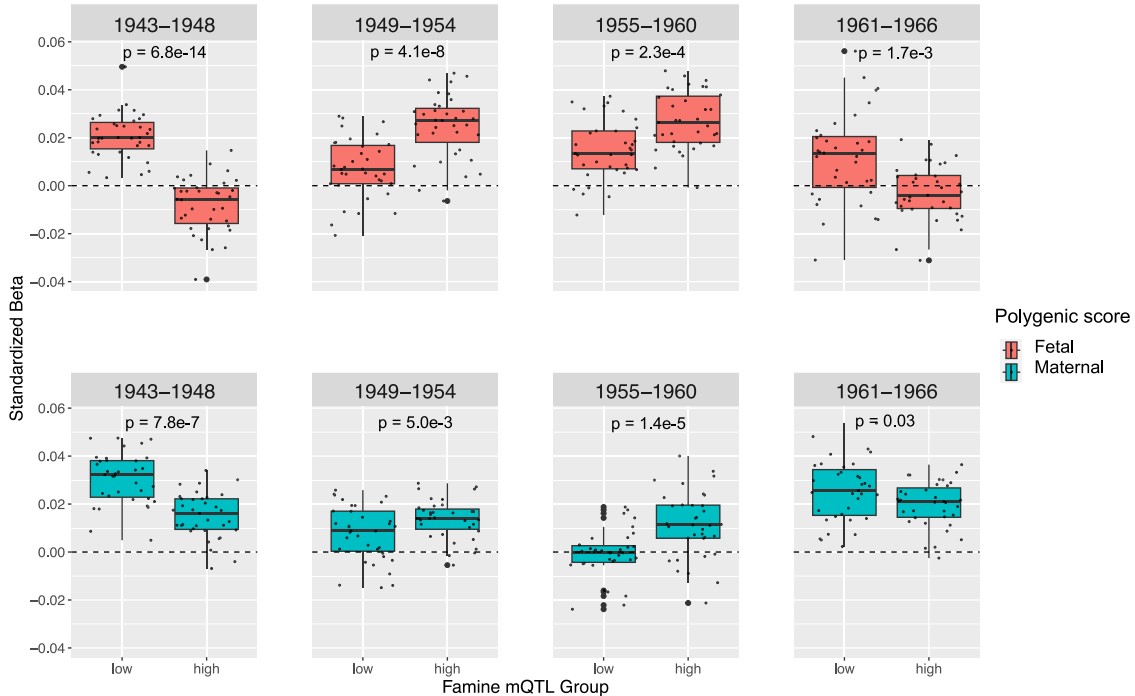

**Fig. 4 | Associations between polygenic scores (PGSs) for fetal and maternal effects on birthweight and cortical surface area as a function of a polygenic score for the epigenetic response to famine.** The two-sided multiple linear regression standardized effect sizes (β) were computed, between PGSs for fetal (red) and maternal (cyan) effects on birthweight and cortical surface area, and evaluated as a function of a polygenic score for the epigenetic response to famine. The analyses were conducted in the following birth year periods: (high famine: $n = 3424$, low famine: $n = 3464$), 1949–1954 (high: $n = 4001$, low: $n = 4047$), 1955-1960 (high: $n = 3179$, low: $n = 3170$), and 1961–1966 (high: $n = 2503$, low: $n = 2468$). Two-sided paired samples t-tests were used to compare profiles of standardized betas between groups. Each box plot depicts the following: the center is the median, the minima and maxima are the bottom and top of the whiskers, respectively, and the lower and upper bounds of the box is the 1st and 3rd quartiles, respectively.

### A. Maternal genes

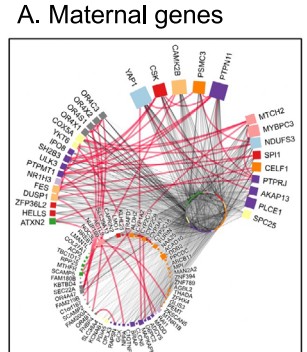

### B. Fetal genes

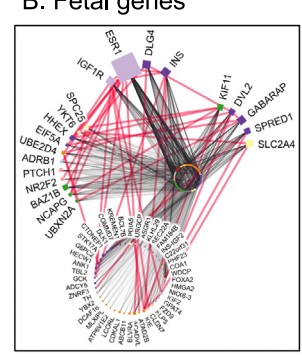

### C. Famine genes

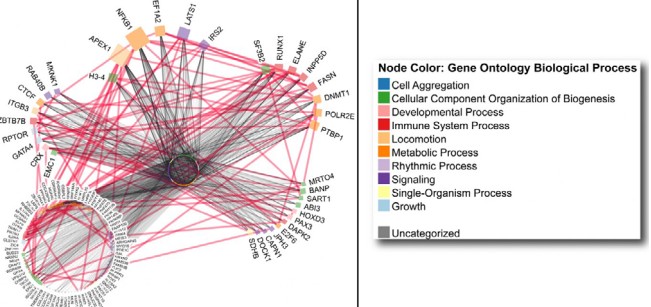

**Fig. 5 | Protein–protein interaction networks for maternal, fetal, and famine genes.** Deregulated (**A**) maternal, (**B**) fetal, or (**C**) famine genes were used as a query to obtain physical protein interactions from Integrated Interactions Database. For simplicity, self-interactions were removed. Node color represents Gene Ontology Biological Process. Only fetal/maternal/famine genes within the network have visible name, remaining genes are organized as circle in the middle. Node size corresponds to the number of interacting partners in the network. Red edges highlight direct interactions among the index (i.e., fetal/maternal/famine) genes; all other interactions are rendered as gray edges. Networks were analyzed and visualized in NAViGaTOR.

### Exploring the biology of the maternal, fetal, and famine genes

To gain insights into the possible mechanistic pathways involved in the observed associations between the maternal (and fetal) genetic variants and the tangential growth of the cerebral cortex, and their modulation by famine-related genetic variants, we have carried out several bioinformatics-based analyses.

First, we identified all protein–protein interactions involving protein-coding genes associated with maternal, fetal, and famine-related genetic variants. Using FUMA[30], maternal, fetal, and famine-related variants ($p < 5 \times 10^{-8}$) mapped onto 98, 71, and 128 genes,

respectively, henceforth termed "maternal", "fetal" and "famine" genes. Note that—at the gene level—there is minimal overlap between the three sets of genes (8 out of the total of 158 fetal and maternal genes, 0 for the famine genes; Fig. S11). Furthermore, only PGSs for the overall birthweight and its fetal component show correlations with PGS of surface area; maternal and famine PGSs do not correlate with the surface-area PGS (Fig. S12).

For the *maternal genes*, we observed a number of clusters of interacting proteins coded by the 98 genes (Fig. 5A), with their identified protein-protein interactions falling into a number of biological

processes. The most connected proteins, namely YAP1 (Yes Associated Protein 1; a part of the Hippo signaling pathway), PTPN11 (Protein Tyrosine Phosphatase Non-Receptor Type 11; a member of the protein tyrosine phosphatase family involved in cell growth), and CSK (C-Terminal Src Kinase; regulation of cell growth, differentiation, migration, and immune response), are associated with multiple biological processes involved in growth, signaling and immune system, respectively. Furthermore, many of these proteins show interactions with several members of the Cytochromes P450 superfamily (CYP1A2, CYP2C9, CYP2C18, CYP2C19, CYP3A7), which are involved oxidation of steroids, fatty acids, and xenobiotics (circle in the center of this network). Gene Ontology (GO) analysis of genes/proteins interacting within the network (red lines in Fig. 5A) identified enrichment in a number of biological processes and molecular functions, with the top pathways involving several P450 pathways and a number of specific catabolic and metabolic processes (Supplementary Data 1). Note that, in addition to these within-network interactions, there are many protein-protein interactions between the maternal and famine networks (Fig. S12).

For the *fetal genes*, we observed a highly interconnected network of 71 proteins (Fig. 5B) dominated by those involved in insulin signaling (INS [insulin], IGF1R [Insulin Like Growth Factor 1 Receptor] and IGF2 [Insulin Like Growth Factor 2) and steroid hormones (ESR1 [Estrogen Receptor 1]). Gene Ontology (GO) analysis of genes/proteins interacting within the network (red lines in Fig. 5B) identified enrichment in a number of biological processes and molecular functions, with the top pathways involving insulin and glucose homeostasis (Supplementary Data 1). There are also many protein-protein interactions between the fetal and famine networks (Fig. S13).

For the *famine genes*, the network of 128 proteins (Fig. 5C) was dominated by within-network interactions of several proteins involved in DNA repair and tumor suppression (APEX1 [Apurinic/Apyrimidinic Endodeoxyribonuclease 1], LATS1 [Large Tumor Suppressor Kinase 1]), gene transcription and regulation of gene expression (POLR2E [RNA Polymerase II, I And III Subunit E], PTBP1 [Polypyrimidine Tract Binding Protein 1], GATA4 [GATA Binding Protein 4], CTCF [CCCTC-Binding Factor], DNMT1 [DNA Methyltransferase 1), as well as the transcriptional response to stimuli such as cytokines, oxidant-free radicals, and bacterial or viral products (NFKB1 [Nuclear Factor Kappa B Subunit 1]). Gene Ontology (GO) analysis of genes/proteins interacting within the network (red lines in Fig. 5C) identified enrichment in a number of biological processes and molecular functions, with the top pathways involving DNA-templated and RNA-polymerase-based regulation of transcription, as well as a small number of cellular components, including intracellular organelles and the nucleus (Supplementary Data 1). As mentioned above, there are also many protein-protein interactions between the famine network and each of the two birthweight-related networks (i.e., maternal and fetal; Fig. S13).

## Discussion

Our observations confirm the expected phenotypic relationship between an overall growth of the fetus, indexed by birthweight, and the tangential growth of the cerebral cortex, indexed by surface area[15–17]. Albeit subtle ( ~ 4% variance explained), it points to shared mechanistic pathways between the two. Subsequent analyses using two sets of genetic variants, namely those influencing fetal growth via maternal and fetal genomes, respectively[12], revealed striking differences in the role of these two sets of genes as a function of the year of birth. Individuals born during and shortly after World War II (WWII) stood out in that their "maternal" but not "fetal" genes related to cortical growth. We speculate that this positive relationship between the "maternal" genes and cortical growth was brought about by an unfavorable environment during this period. This explanation is supported by an instrumental variable that mimics, via

a set of genetic variations, the known epigenetic response to famine during WWII.

Previous studies suggest that genetic contributions to a particular trait may vary as a function of environment. For instance, in several bird species, heritability of growth appears to be lower under less favorable conditions, including poor nutrition[31]. Some studies also suggest that different sets of genes may influence growth under more or less favorable conditions, respectively[32]. Conceptually, our results are consistent with these findings. The two sets of birth-related genes differ in their primary functions. Thus, while fetal genes involve processes directly related to organ growth, including insulin signaling, maternal genes appear to have more general functions, including cell proliferation, differentiation, and migration, as well as immune signaling. Importantly, the latter—maternal—gene set also includes a large number of enzymes (Cytochrome P450 [CYP] superfamily) involved in metabolism (detoxification) of endogenous and exogenous chemicals[33]. In experimental models, food restriction during gestations increases activity of CYP enzymes[34]. Furthermore, levels of xenobiotics entering the circulation may vary as a function of the host's microbiome[35]; hence, further importance of these enzymes in the context of possible changes in the microbiome due to major dietary shifts, such as those that occurred during WWII[21,22].

Of course, we do not know what happened to the UK Biobank participants who (likely) experienced, *in utero*, food restriction during and shortly after WWII. But the protein-protein interaction network based on the famine PGS offers some clues. These "famine" genes are implicated in DNA repair, tumor suppression, and transcriptional regulation, in particular in response to cellular stress and immune activation. We suggest that the "maternal" genes (involved in cell proliferation and detoxification) are in an excellent position to counteract such a molecular adversity. Furthermore, our results suggest that this hypothesized compensatory action has its limits: when the real exposure (birth year) combines with the virtual exposure (famine PGS), the "maternal" genes are no longer capable of diminishing its impact on cortical growth.

This report is limited by its observational nature, the small sample of the Dutch famine methylation study, and the lack of measurements obtained during gestation and infancy of the UK Biobank participants. Despite the small sample of the Dutch famine methylation study and limited evidence of transgenerational effects, we felt that the similarities between the 1943–48 and 1961–66 cohorts warranted exploring epigenetic mechanisms as one possible—albeit tentative—explanation. As expected, among the participants born in 1961-1966, two-thirds of their mothers were born outside the WWII period. Therefore, a potential transgenerational transmission from participants whose mothers were exposed to WWII alone cannot explain the findings in the 1961–1966 cohorts, which must be treated as tentative. Moreover, although there is some overlap between the birthweight GWAS samples and the polygenic score sample, we have conducted sensitivity analyses demonstrating that this is unlikely to affect the PGS weights, and consequently, the findings (Fig. S14-15). Furthermore, while the maternal genetic variant effects were originally identified in mothers, we have used them as proxies of maternal genotypes given that 50% of the genetics are shared between mothers and their offspring. As such, the maternal effect PGSs in the offspring (UKBB participants in this case) serve as indirect markers of their uteroplacental environment. The latter is consistent with our observation that maternal PGS does not correlate with the SA PGS, while the fetal PGS does (see Fig. S10). Altogether, this study has revealed robust patterns of associations between two sets of genetic variations underpinning fetal growth and the tangential growth of the human cerebral cortex, and their variations by the year of birth. Moreover, the introduction of an instrumental variable modeling the known effects of famine on epigenome provides an independent validation of the past (global) events. This instrumental variable also opens new opportunities for studying gene-

environment interactions underlying various neurodevelopmental disorders. Finally, this work shows clearly that food security is a *sine qua* non-condition for healthy brain development.

## Methods

### Participants

The UK Biobank is a genotyped and richly phenotyped cohort of over 500,000 participants, recruited between 2006 and 2010 and assessed in 22 centers in the United Kingdom[13]. All participants provided electronic signed informed consent, underwent interviews, physical assessments, and provided blood, urine, and saliva samples[13]. The phenotypic assessments included anthropometric measures, multimodal imaging, accelerometry, biochemical assays, questionnaires, and health diagnostics[13]. The UK Biobank study was approved by the North West Multi-center Research Ethics Committee as a Research Tissue Bank (see: https://www.ukbiobank.ac.uk/learn-more-about-uk-biobank/about-us/ethics). This study was approved under the UK Biobank Resource Application Number 43688 and by local ethics committees at the Research Institute of the Hospital for Sick Children (SickKids) and the Centre Hospitalier Universitaire (CHU) Sainte-Justine.

### Cortical surface area

Multimodal MRI scans, including T1-weighted images, were acquired in a subset of UK Biobank participants ($n = 39,501$) using a Skyra 3 T scanner with a 32-channel head coil[36]. Among the participants with MRI data, those with born outside the UK ($n = 2835$), less than 2500 g ($n = 2007$), and/or as part of multiple births ($n = 1274$), were excluded, leaving 33,900 participants. Regional and global values of SA were extracted using FreeSurfer based on the Desikan-Killiany atlas[14]. Left and right SA values were summed. Regional SA values were not adjusted for total SA as done previously[7,37,38].

### Polygenic scores

Polygenic scores were calculated for: (i) own birthweight, (ii) fetal birthweight variants, (iii) maternal birthweight variants, (iv) regional cortical SA, and (v) famine (see Famine PGS). Summary statistics for birthweight and its two components were obtained from the EGG consortium and the UK Biobank[12]. Summary statistics for global and regional SA were obtained from the Enhancing NeuroImaging Genetics through Meta-Analysis (ENIGMA) and UK Biobank cohorts[6], without adjustment for total SA.

Prior to generating polygenic scores, the participants and SNPs were quality controlled (QC) in a sex-specific manner. Participants without genetic information or genetic sex not available ($n = 14,242$), heterozygosity or missingness outliers ($n = 963$), a mismatch between genetic and reported sex ($n = 197$), sex chromosomal aneuploidy ($n = 651$), and non-European ancestry ($n = 78,290$; field 22006) were excluded. Additionally, participants with more than ten 3rd-degree relatives ($n = 161$) or not in kinship analyses ($n = 2$) were excluded, followed by the sex-specific removal of individuals with close kinship using the R package 'ukbtools' version 0.11.3 (KING coefficient = 0.0884), excluding 11,162 females and 6043 males[39]. Quality control at the level of SNPs comprised exclusions based on > 5% missingness, duplicated SNPs, a minor allele frequency <0.01, a significant deviation from Hardy Weinberg Equilibrium (threshold: $p < 1 \times 10^{-10}$), or an INFO score <0.8, as done previously[40]. Following the genetic QC, the final "genetic" dataset included 209,383 females with 8,637,896 SNPs, and 181,389 males with 8,639,777 SNPs. Following this, there were 14,905 females and 14,142 males with values for cortical SA.

Using the above summary statistics as base datasets, PRSice-2[18,41] was implemented to derive polygenic scores. Only genome-wide significant ($p < 5 \times 10^{-8}$) and independent (clumping: $r^2 \geq 0.1$, distance = 250 kb) SNPs were retained. The maternal PGS contained 39 SNPs in both sexes while the fetal PGS contained 34 SNPs in males and 35 SNPs in females. The famine PGS contained 188 SNPs in males and 189 in females. The global surface-area PGS contained 21 SNPs in both sexes while the regional surface-area PGSs comprised a median of 13 SNPs per region in both sexes.

After individual-level polygenic scores were computed using PRSice-2, multiple linear regressions, adjusting for age at MRI, sex, and the first 10 principal components of genetic ancestry, were performed between each birthweight polygenic score and cortical SA for the 34 regions and globally. We have also evaluated correlations between the maternal, fetal and famine PGSs and the surface-area PGS. We computed FDR-adjusted $p$-values for each test of 35 $p$-values (i.e., 34 regions and globally). Paired samples t-tests were used to compare profiles of effect sizes (standardized betas) between groups. Standardized betas were computed by standardizing the continuous variables before fitting the models (mean = 0, standard deviation = 1). Statistical analyses were conducted and figures were generated using R version 4.1.1 and 'tidyverse' version 1.3.2 functions[42,43].

### Famine polygenic score

In an investigation of the Dutch famine cohort comparing methylation status of 24 famine-exposed cases and their same-sex siblings, CpGs were grouped according to genomic annotations, and the authors identified that five annotations demonstrated a significant difference in methylation between cases and controls[28]. Within those annotations, differentially methylated regions (DMRs; $p_{FDR} < 0.05$) were identified, among which 181 had significantly different methylation between cases and controls. Critically, the degree of methylation between adjacent CpGs is highly correlated[44]. Thus, we extracted the positional ranges of the 181 DMRs[28], transformed from genome build hg18 to hg19 using the UCSC LiftOver (https://genome.ucsc.edu/cgi-bin/hgLiftOver)[45]. Within these 181 DMRs, 538 unique CpGs were contained in the Illumina Infinium HumanMethylation450 (450 K) BeadChip array. Next, CpGs were mapped to methylation quantitative trait loci (mQTLs) using the GoDMC Atlas version 0.1.0 (2019-11-01)[46], identifying 446 clumped, genome-wide significant ($p < 5e-8$) mQTLs. Following this, indels and duplicate mQTLs were removed, retaining the mQTL within a duplicate set with the smallest p-value or the largest absolute effect size for duplicate mQTLs with $p = 0$, resulting in 377 unique mQTLs (346 cis, 31 trans). Finally, following matching for SNPs in our quality-controlled UK Biobank target dataset, removing ambiguous SNPs and clumping across SNPs ($r^2 = 0.1$; kb = 250), we retained 189 SNPs in females and 188 SNPs in males for further analysis (Fig. S6A). In order to compute the famine polygenic scores, the effect sizes from the GoDMC mQTL atlas were weighted by the directionality of the effects from the DMRs reported by ref. 28 (+1: hypermethylation among famine-exposed, relative to controls; −1: hypomethylation among famine-exposed, relative to controls). The calculation of famine PGSs is illustrated in Fig. S6B.

### Protein–protein interactions

Physical protein–protein interactions for the deregulated fetal, maternal and famine genes were obtained from the Integrated Interactions Database (IID) version 2021-05 (https://ophid.utoronto.ca/iid)[47]. We included all interaction sources; the full list of interactions is provided in Supplementary Data 2. Enrichment analyses were performed using Enrichr (https://maayanlab.cloud/Enrichr) and two-tailed hypergeometric tests, with p-value adjustment using the Benjamini-Hochberg method. For the maternal genes, we have submitted 98 genes into IID, of which 94 had protein interactions. The resulting network comprised a total of 7007 proteins and 14,505 protein–protein interactions, with the 94 maternal proteins being directly connected by 61 interactions. Other interactors included 6913 proteins with 43 interactions among them (middle circle in Fig. 5A). For the fetal genes, we have submitted 71 genes into IID, of which 64 had protein interactions. The resulting network comprised a total of 7539

proteins and 14,102 protein-protein interactions, with the 64 fetal proteins being directly connected by 41 interactions. Other interactors included 7475 proteins with no direct interactions (middle circle in Fig. 5B). For the famine genes, we have submitted 128 genes into IID, of which 122 proteins had protein interactions. The resulting network comprised a total of 8457 proteins and 20,162 protein–protein interactions, with the 122 famine proteins being directly connected by 93 interactions. Other interactors included 8335 proteins with two 2 interactions among them (middle circle in Fig. 5C). Individual protein interaction networks were then loaded into NAViGaTOR[48] ver. 3.0.17 for further annotation, analysis, and visualization. The final network was exported in SVG file format, and finalized with legends in Adobe Illustrator ver. 27.2.

## Reporting summary
Further information on research design is available in the Nature Portfolio Reporting Summary linked to this article.

## Data availability
The data can be provided by the UK Biobank pending scientific review and a completed material transfer agreement. Applications for access to the data can be completed at: https://www.ukbiobank.ac.uk/enable-your-research/apply-for-access. Physical protein-protein interactions were obtained from the Integrated Interactions Database (IID) version 2021-05 (https://ophid.utoronto.ca/iid).

## Code availability
The custom R code for selecting the mQTLs for the famine PGS is provided in the supplementary information (Supplementary Software 1).

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

## Acknowledgements

We would like to thank Dr. Zhijie Liao for her helpful suggestions and Dr. Jean Shin and Andrei Mouraviev for technical assistance. Canadian Institutes of Health Research (TP, DEV [Postdoctoral scholarship: MFE - 187903]) National Institutes for Health (ZP) UK Biobank Resource under Application Number 43688.

## Author contributions

Conceptualization: DEV, IJ, ZP, TP. Methodology: DEV, IJ, ZP, TP. Visualization: DEV, IJ. Funding acquisition: ZP, TP. Project administration: ZP, TP. Supervision: TP Writing—original draft: DEV, IJ, TP. Writing—review & editing: DEV, IJ, ZP, TP.

## Competing interests

The authors declare no competing interests.
