## [Peer Review File · Nature Communications]

Intrauterine growth and the tangential expansion of the human cerebral cortex in times of food scarcity and abundanceREVIEWER COMMENTS

Reviewer #1 (Remarks to the Author):

Vosberg and colleagues used the UK Biobank resource to investigate the relationship between birthweight, polygenic scores (PGSs) associated with birthweight and intra-uterine conditions, and measures of sizes of brain regions obtained using imaging (i.e. cortical surface areas). Among >32k individuals, they observed associations between PGSs of overall birthweight, of fetal influences on birthweight, of maternal influences on birthweight, and of genetic variants that correlate with differential methylation previously observed to be associated with prenatal exposure to the Dutch Famine. Based on the pattern of associations (in different birth cohorts, between men and women, interaction tests), the authors conclude that adverse prenatal conditions influence the impact of biological processes marked by the various PGSs.

The research question is of potential interest and the conclusion intriguing, the interpretation has various gaps that are inevitable for the research approach and it remains unclear if specific conclusions are warranted or that the work should be regarded as hypothesis-generating.

1. The association between birthweight and the size of brain regions is as expected. Larger individuals also have larger heads and brains. Indeed, the association was observed before. PGSs are limited in establishing specific, non-pleiotropic associations. For example, it should be tested whether PGSs for adult height (with which birthweight PGSs are known to overlap) show similar associations.
2. The authors apply maternal and fetal PGSs for birthweight. The maternal PGS was established as genetic variants in the mother affecting birthweight of their offspring. In this study, the maternal PGS appears to be applied to the offspring itself. This is not logical and results from such analyses (including differences between various PGSs) have no (clear) interpretation and were better not reported.
3. The authors should test for pleiotropy using the methods from the field of Mendelian randomization (Egger regression and Cochran's Q test) and also should perform multivariable MR (correcting for related PGSs and factors).
4. The construction of a PGS for prenatal famine-associated differences in DNA methylation serves an interesting goal but may be stretching possibilities. The famine-associated methylation differences relate to regions and encompass many CpGs not covered by methylation arrays that were used as a resource to generate PGS. What empirical evidence do the authors have that the cis-meQTLs they selected affected the whole region? It would be of interest if the authors specifically test the CpGs that were measured in the famine study that overlap with those on the array. There have also been reports of the famine cohort using the same array as used for the meQTL resource. This may be a more direct approach. I understood that the prenatal famine-associated differences in DNA methylation were specific for early gestation exposure while the authors seem to write that later periods in gestation are particularly relevant for their outcome of interest.
5. The authors mention that there would be evidence for transgenerational effects from Famine studies. The existence of transgenerational (epigenetic) effects is controversial at least. The work cited by the authors to support their claim is a single paper on birthweight based on a very limited sample size.

Reviewer #2 (Remarks to the Author):

Using data from the UK Biobank, the authors found that for individuals born 1942-1948 & 1961-1966 the maternal but not the fetal birthweight PGS influenced SA. 1943-1948 driven by males. For individuals born 1949-1960 the fetal but not the maternal PGS predicted SA. They hypothesised 1942-1948 results due to food restriction and 1961-1966 results due to intergenerational transmission of epigenetic responses to food scarcity. To assess this, they built a PGS to index methylation differences seen in those who experienced the Dutch Hunger Winter famine.

These results are interesting and intriguing and there are a number of suggestions to which would allow the authors to refine these results somewhat.

Major issue

The Warrington et al birthweight GWAS paper on which the birthweight PGS is built included UK Biobank participants in the analyses. Did the authors obtain leave-one-out summary stats for this GWAS?

Similarly, while leave-one-out UK Biobank summary stats are available for the Grasby et al ENIGMA paper the authors do not report using these.

If the summary stats used to build the PGS included UK Biobank participants, then the effects of these PGS analyses are subject to overfitting.

Specific comments

1) Page 3 split off this cause to be a separate sentence & restructure if so it is easier to understand ionizing radiation of the (monkey) fetus during early gestation reduces surface area (SA) (3).

2) The assumptions around the effects seen in the 1961-1966 age group being due to rationing are based several assumptions, including that their mothers were born between 1940-1948, to grandmothers living in the UK (or Europe) during this period.

This would mean that the mothers are assumed to have been aged between 13 and 23 when the UK participant was born. Given that participants reported their mothers age (item 1845) and these data are available for 199,258 UK Biobank participants it would be good to see if the effect of the famine PGS holds for the participants whose mothers were in fact born during rationing. If the theory is correct one would expect this effect to become stronger among this subset of participants.

3) Can the authors please clarify that they have not included UK Biobank participants born as part of a multiple birth (item 1777) Born prematurely or born outside the UK (item 20115) in their analyses.

4) Given the impact of food rationing differed between urban and rural setting in the UK and the UK Biobank did not recruit from the London or some of the other major cities where the food scarcity was particularly severe among lower income families can the authors examine whether the famine PGS effects differ by place of birth (items 129 and 130)?

5) Are these same patterns of effects seen between famine PGS and birthweight for the 4 different birth-year groupings

6) Although there methods in the supplementary materials it would be useful to add a few methodological details in the paper such as the imaging measures were extracted from FreeSurfer using the DK atlas and summed across the 2 hemispheres and that PGS were calculated with PGSice-2 (rather than SBayesR or one of the other methods).

7) Why did the authors chose to only use genome-wide significant SNPs in the PGS calculation rather than building the most powerful predictor possible?

8) Can the authors please be more explicit about the limitations of this work and the assumptions that have been made?

REVIEWER COMMENTS

Reviewer #1 (Remarks to the Author):

Vosberg and colleagues used the UK Biobank resource to investigate the relationship between birthweight, polygenic scores (PGSs) associated with birthweight and intra-uterine conditions, and measures of sizes of brain regions obtained using imaging (i.e. cortical surface areas). Among >32k individuals, they observed associations between PGSs of overall birthweight, of fetal influences on birthweight, of maternal influences on birthweight, and of genetic variants that correlate with differential methylation previously observed to be associated with prenatal exposure to the Dutch Famine. Based on the pattern of associations (in different birth cohorts, between men and women, interaction tests), the authors conclude that adverse prenatal conditions influence the impact of biological processes marked by the various PGSs. The research question is of potential interest and the conclusion intriguing, the interpretation has various gaps that are inevitable for the research approach and it remains unclear if specific conclusions are warranted or that the work should be regarded as hypothesis-generating.

Comment R1.1. The association between birthweight and the size of brain regions is as expected. Larger individuals also have larger heads and brains. Indeed, the association was observed before. PGSs are limited in establishing specific, non-pleiotropic associations. For example, it should be tested whether PGSs for adult height (with which birthweight PGSs are known to overlap) show similar associations.

Response R1.1. In order to address this question, we have computed PGSs for height, using the most recent summary statistics from the GIANT consortium ¹, and used the height PGS as a covariate in our analyses. Firstly, the correlations between the height and birthweight PGSs were significant but accounted for very little of the variance among females (fetal: $r^2 = 0.003$, $p = 2.89e-10$; maternal: $r^2 = 0.003$, $p = 1.32e-09$) and males (fetal: $r^2 = 0.002$, $p = 2.00e-07$; maternal: $r^2 = 0.004$, $p = 4.58e-11$). Next, with the height PGS as a covariate in our models assessing the associations between birthweight (fetal; maternal) PGSs and SA, the results were largely unchanged, as demonstrated in the figure below.

We have added the following information in the Results:

“Additionally, given the known relationship between height and head circumference¹⁹, we have examined the correlations between height PGS (using the GIANT Consortium¹) and birthweight PGSs (fetal and maternal). We have also re-analysed the correlations between the cortical surface area and birthweights PGSs after adjusting the latter for the height PGS. These results are provided in the Supplementary Material (Fig. S2). The findings remained the same after this adjustment”.

Comment R1.2. The authors apply maternal and fetal PGSs for birthweight. The maternal PGS was established as genetic variants in the mother affecting birthweight of their offspring. In this study, the maternal PGS appears to be applied to the offspring itself. This is not logical and results from such analyses (including differences between various PGSs) have no (clear) interpretation and were better not reported.

Response R1.2. While the maternal genetic variant effects were indeed identified in mothers, we believe it is valid to use them as proxies of maternal genotypes given that 50% of the genetics are shared between mothers and their offspring. As such, the maternal effect PGSs in the offspring (UKBB participants in this case) serve as indirect markers of their uteroplacental environment. The latter is consistent with our observation that maternal PGS does not correlate with the SA PGS, while the fetal PGS does (see Figure S10).

We have added this information in the Discussion:

“Additionally, while the maternal genetic variant effects were identified in mothers, we have used them as proxies of maternal genotypes given that 50% of the genetics are shared between mothers and their offspring. As such, the maternal effect PGSs in the offspring (UKBB participants in this case) serve as indirect markers of their uteroplacental environment. The latter is consistent with our observation that maternal PGS does not correlate with the SA PGS, while the fetal PGS does (see Figure S10).”

Comment R1.3. The authors should test for pleiotropy using the methods from the field of Mendelian randomization (Egger regression and Cochran’s Q test) and also should perform multivariable MR (correcting for related PGSs and factors).

Response R1.3. To address this suggestion, we conducted two-sample Mendelian Randomization using the R package, TwoSampleMR, between the birthweight genetic exposures (fetal effect, maternal effect) and cortical SA. We did not identify evidence of causal effects of birthweight on surface area (all $p \geq 0.10$). Moreover, conducting multivariable MR (correcting for height genetics), the associations remained non-significant (all $p \geq 0.10$). Applying MR Egger on these genetic instruments, there was no evidence of directional pleiotropy for the fetal ($p = 0.51$) or maternal ($p = 0.22$) effects. Moreover, as addressed in R1.1, including a polygenic score for height in our model did not alter our findings.

Comment R1.4. The construction of a PGS for prenatal famine-associated differences in DNA methylation serves an interesting goal but may be stretching possibilities. The famine-associated methylation differences relate to regions and encompass many CpGs not covered by methylation arrays that were used as a resource to generate PGS. What empirical evidence do the authors have that the cis-meQTLs they selected affected the whole region? It would be of interest if the authors specifically test the CpGs that were measured in the famine study that overlap with those on the array. There have also been reports of the famine cohort using the

same array as used for the meQTL resource. This may be a more direct approach. I understood that the prenatal famine-associated differences in DNA methylation were specific for early gestation exposure while the authors seems to write that later periods in gestation are particularly relevant for their outcome of interest.

Response R1.4. First of all, we agree with the reviewer that this is "...an interesting goal but may be stretching possibilities". We have pursued it only because the similarities between the 1943-48 and 1961-66 cohorts are so striking, and we felt these warranted exploring epigenetic mechanisms as one possible – albeit tentative – explanation. We have added the following sentence to this effect in the Discussion:

"Despite the small sample of the Dutch famine methylation study and limited evidence of transgenerational effects, we felt that the similarities between the 1943-48 and 1961-66 cohorts warranted exploring epigenetic mechanisms as one possible – albeit tentative – explanation."

As for the methodological issues raised by the reviewer, we contacted the authors of the famine study, Tobi et al. (2014), to request the specific CpGs tested within the differentially methylated regions. We were informed that these were not available since the analyses predated CpG identifiers. Nevertheless, the CpGs we selected were specifically those that overlapped with the differentially methylated regions. Rather than test all available CpGs, Tobi et al. (2014) annotated their CpGs according to genomic annotations and identified that five annotations demonstrated a significant difference in methylation between cases and controls. Within those annotations, differentially methylated regions were identified, among which 181 had significantly different methylation between cases and controls. Critically, the degree of methylation between adjacent CpGs is highly correlated ².

We have added this information in the Methods as follows:

"In an investigation of the Dutch famine cohort comparing methylation status of 24 famine-exposed cases and their same-sex siblings, CpGs were grouped according to genomic annotations and the authors identified that five annotations demonstrated a significant difference in methylation between cases and controls ²⁸. Within those annotations, differentially methylated regions (DMRs; $p_{FDR} < 0.05$) were identified, among which 181 had significantly different methylation between cases and controls. Critically, the degree of methylation between adjacent CpGs is highly correlated ⁴⁴."

Comment R1.5. The authors mention that there would be evidence for transgenerational effects from Famine studies. The existence of transgenerational (epigenetic) effects is controversial at least. The work cited by the authors to support their claim is a single paper on birthweight based on a very limited sample size.

Response R1.5. We have added this limitation to our Discussion section (see Response R2.9.) below.

Reviewer #2 (Remarks to the Author):

Using data from the UK Biobank, the authors found that for individuals born 1943-1948 & 1961-1966 the maternal but not the fetal birthweight PGS influenced SA. 1943-1948 driven by males. For individuals born 1949-1960 the fetal but not the maternal PGS predicted SA. They hypothesised 1942-1948 results due to food restriction and 1961-1966 results due to intergenerational transmission of epigenetic responses to food scarcity. To assess this, they

built a PGS to index methylation differences seen in those who experienced the Dutch Hunger Winter famine.

These results are interesting and intriguing and there are a number of suggestions to which would allow the authors to refine these results somewhat.

Comment R2.1. The Warrington et al birthweight GWAS paper on which the birthweight PGS is built included UK Biobank participants in the analyses. Did the authors obtain leave-one-out summary stats for this GWAS? Similarly, while leave-one-out UK Biobank summary stats are available for the Grasby et al ENIGMA paper the authors do not report using these. If the summary stats used to build the PGS included UK Biobank participants, then the effects of these PGS analyses are subject to overfitting.

Response R2.1. In order to address this important comment, we contacted the authors of Warrington et al. (2019), and obtained summary GWAS statistics for birthweight based on the EGG Consortium alone, *excluding* the UK Biobank. Summary GWAS statistics for the maternal and fetal effects, excluding the UK Biobank cohort, were not available. Thus, using the birthweight GWAS of the EGG cohort alone, we filtered for the 35 “fetal effect” and 39 “maternal effect” SNPs, and identified that these were highly correlated between the original cohort (EGG and UK Biobank) and EGG cohort alone (fetal: $r = 0.99$, $p = 1.32e-27$; maternal: $r = 0.90$, $p = 5.06e-15$), as demonstrated in the figures below.

Moreover, although both our base and target samples contain UK Biobank participants, our target sample is filtered for those with MRI data ($n = \sim 32k$), representing $\sim 14\%$ of the total UK Biobank sample used in the GWASs by Warrington et al. ($n = \sim 220k$). Thus, we conducted sex-specific GWASs of birthweight for the total sample in the UK Biobank and for the sample excluding those with MRI data, and assessed the correlations of the effect sizes of the SNPs between the two samples. As indicated in the following plots, the effect sizes between the two samples were very highly correlated (all $r \geq 0.99$) for both the fetal and maternal SNPs.

Females

Fetal effect on birthweight (35 SNPs)

Maternal effect on birthweight (39 SNPs)

Males

Fetal effect on birthweight (35 SNPs)

Maternal effect on birthweight (39 SNPs)

Regarding the genetics of cortical surface area, we compared the effect sizes for the 21 SNPs between the summary statistics derived from the sample with ($n = 32k$) and without the UK Biobank cohort ($n = 22k$). Again, the correlation was very high ($r > 0.99$).

We provide the above analyses in the Supplementary Material.

Comment R2.2. Page 3 split off this cause to be a separate sentence & restructure if so it is easier to understand ionizing radiation of the (monkey) fetus during early gestation reduces surface area (SA) (3).

Response R2.2. This section has been rewritten into two sentences, as indicated below.

“In primates, the phase of *symmetric division of progenitor cells* in the proliferative zones during the first trimester is particularly important for the tangential growth through the additions of ontogenetic columns³. Ionizing radiation of the (monkey) fetus during early gestation reduces surface area (SA) of the cerebral cortex⁴.”

Comment R2.3. The assumptions around the effects seen in the 1961-1966 age group being due to rationing are based several assumptions, including that their mothers were born between 1940-1948, to grandmothers living in the UK (or Europe) during this period. This would mean that the mothers are assumed to have been aged between 13 and 23 when the UK participant was born. Given that participants reported their mothers age (item 1845) and these data are available for 199,258 UK Biobank participants it would be good to see if the effect of the famine PGS holds for the participants whose mothers were in fact born during rationing. If the theory is correct one would expect this effect to become stronger among this subset of participants.

Response R2.3. Among the UK Biobank participants born in 1961-1966, their mothers were born between 1913-1950 (1st quartile: 1933, median: 1937, 3rd quartile: 1940). Thus, about a quarter of the mothers of the 1961-1966 participants were born during WWII. Moreover, a

subset of the older mothers of these participants were born during WWI, while the remainder were born during the interwar period. While our findings may indicate an intergenerational mechanism, we have reiterated in the limitations section of our Discussion that this must be treated tentatively, as described in the response R2.9 below.

Comment R2.4. Can the authors please clarify that they have not included UK Biobank participants born as part of a multiple birth (item 1777) Born prematurely or born outside the UK (item 20115) in their analyses.

Response R2.4. In order to address this, we have now excluded those born as part of a multiple birth and those born outside the UK. Moreover, since gestational duration was not available for the UK Biobank participants, we instead excluded those with a birthweight of less than 2,500 g, following the example of Warrington et al. (2019). With our updated exclusion criteria, the findings were essentially unchanged, as demonstrated below. The main text and figures have been updated accordingly.

Fig. 1

Fig. 2

Fig. 3

Fig. 4

Comment R2.5. Given the impact of food rationing differed between urban and rural setting in the UK and the UK Biobank did not recruit from the London or some of the other major cities where the food scarcity was particularly severe among lower income families can the authors examine whether the famine PGS effects differ by place of birth (items 129 and 130)?

Response R2.5. In order to address this comment, we used the north and east coordinates of place of birth for each participant to identify the geospatial administrative unit they were born in, namely the Middle Layer Super Output Areas (MSOA). We then used the urban/rural classification for each MSOA provided by the UK's Office for National Statistics. Among participants with MRI and genetic data and passing our inclusion criteria, there were 23,788 participants born in an urban region and 2,573 participants born in a rural region. While the urban subset was very similar to our overall analysis, there were fewer significant associations among the substantially smaller rural subset, likely due to a lack of statistical power.

Urban only (n = 23,788)

Rural only (n = 2,573)

Comment R2.6. Are these same patterns of effects seen between famine PGS and birthweight for the 4 different birth-year groupings?

Response R2.6. We conducted associations between the famine PGS and birthweight and did not find any significant effects. These are provided below.

Birth-group	Beta	SE	p
1943-1948	-0.0181	0.0174	0.298
1949-1954	0.00591	0.0145	0.684
1955-1960	-0.0183	0.0152	0.228
1961-1966	-0.00685	0.0165	0.678

Comment R2.7. Although there are methods in the supplementary materials it would be useful to add a few methodological details in the paper such as the imaging measures were extracted from FreeSurfer using the DK atlas and summed across the 2 hemispheres and that PGS were calculated with PGSice-2 (rather than SBayesR or one of the other methods).

Response R2.7. Thank you, these details are now provided in the main text as indicated below.

“The SA values were derived using FreeSurfer and the Desikan-Killiany atlas ⁵ and summed across the two hemispheres.”

“The PGSs were computed at the genome-wide significant threshold using PRSice-2 ⁶.”

Comment R2.8. Why did the authors choose to only use genome-wide significant SNPs in the PGS calculation rather than building the most powerful predictor possible?

Response R2.8. We decided to select the genome-wide significant SNP threshold for several reasons. Firstly, we wanted to focus on SNPs and genes that were most specific and most robustly associated with our traits of interest. Secondly, we sought to use these polygenic scores as instrumental variables and minimize pleiotropic effects, which is relevant to the question raised by Reviewer 1. Finally, given the partial overlap of UK Biobank participants between our base (i.e., GWAS summary statistics) and target (i.e., among whom scores were calculated) samples, restricting our SNPs to GWAS-significant SNPs is likely to minimize overfitting; see also response R2.1.

Comment R2.9. Can the authors please be more explicit about the limitations of this work and the assumptions that have been made?

Response R2.9. We now provide the limitations and assumptions of this work, in the Discussion, as follows.

“This report is limited by its observational nature, the small sample of the Dutch famine methylation study, and the lack of measurements obtained during gestation and infancy of the UK Biobank participants. Despite the small sample of the Dutch famine methylation study and limited evidence of transgenerational effects, we felt that the similarities between the 1943-48 and 1961-66 cohorts warranted exploring epigenetic mechanisms as one possible – albeit tentative – explanation. Importantly, among the UK Biobank participants born in 1961-1966, their mothers were born between 1913-1950 (1st quartile: 1933, median: 1937, 3rd quartile: 1940), indicating that while roughly a quarter were born during WWII, another subset were born during WWI, and the remainder were born during the interwar period. Moreover, although there is some overlap between the birthweight GWAS samples and the polygenic score sample, we have conducted sensitivity analyses demonstrating that this is unlikely to affect the PGS

weights, and consequently, the findings (Fig. S13-14). Furthermore, while the maternal genetic variant effects were originally identified in mothers, we have used them as proxies of maternal genotypes given that 50% of the genetics are shared between mothers and their offspring. As such, the maternal effect PGSs in the offspring (UKBB participants in this case) serve as indirect markers of their uteroplacental environment. The latter is consistent with our observation that maternal PGS does not correlate with the SA PGS, while the fetal PGS does (see Figure S10). Altogether, this study has revealed robust patterns of associations between two sets of genetic variations underpinning fetal growth and the tangential growth of the human cerebral cortex, and their variations by the year of birth.”

References

1. Yengo, L. *et al.* A saturated map of common genetic variants associated with human height. *Nature* **610**, 704–712 (2022).
2. Song, Y., Ren, H. & Lei, J. Collaborations between CpG sites in DNA methylation. *Int. J. Mod. Phys. B* **31**, 1–16 (2017).
3. Rakic, P. Specification of Cerebral Cortical Areas Cortical Neurons Originate Outside the. *Science* **241**, 170–176 (1988).
4. Selemon, L. D. *et al.* Distinct abnormalities of the primate prefrontal cortex caused by ionizing radiation in early or midgestation. *J. Comp. Neurol.* **521**, 1040–1053 (2013).
5. Fischl, B. FreeSurfer. *Neuroimage* **62**, 774–781 (2012).
6. Choi, S. W. & O’Reilly, P. F. PRSice-2: Polygenic Risk Score software for biobank-scale data. *Gigascience* **8**, 1–6 (2019).

REVIEWER COMMENTS

Reviewer #1 (Remarks to the Author):

I thank the authors for their answers and have no further comments.

Reviewer #2 (Remarks to the Author):

The authors have addressed most of the reviewer comments well.

However, I think point 2.3 needs further consideration as the statements being made are quite strong and are likely to be a focus of media attention.

Given so few of the mothers of the 1961-1966 age group were in fact born during the relevant period I think the way these effects are being described and discussed is an over interpretation and needs to be revised. I also think the authors need to provide a sensitivity analysis to show that the effects are stronger among participants whose mothers were born during the relevant period.

I don't think mentioning this in the limitations section is enough.

REVIEWER COMMENTS

Reviewer #1 (Remarks to the Author):

Comment R1.1. I thank the authors for their answers and have no further comments.

Response R1.1. We would like to thank the reviewer for their thoughtful comments, which have improved our manuscript.

Reviewer #2 (Remarks to the Author):

Comment R2.1. The authors have addressed most of the reviewer comments well. However, I think point 2.3 needs further consideration as the statements being made are quite strong and are likely to be a focus of media attention. Given so few of the mothers of the 1961-1966 age group were in fact born during the relevant period I think the way these effects are being described and discussed is an over interpretation and needs to be revised. I also think the authors need to provide a sensitivity analysis to show that the effects are stronger among participants whose mothers were born during the relevant period. I don't think mentioning this in the limitations section is enough.

Response R2.1. We would like to thank the reviewer for their feedback, which has improved our manuscript. Rather than only addressing this point only in the limitations, we now introduce it immediately before proceeding with the famine PGS analyses, and provide a supplementary figure showing the mothers' age distribution, as below. Unfortunately, due to administrative delays, we were unable to conduct the requested sensitivity analysis using *individual-level* data but were able to assess *summary statistics* regarding the ages of the mothers of the participants born in the four different periods, as described below.

“We also reasoned that, given the similarity of these associations between the 1943-1948 and 1961-1966 cohorts, part of this observation might be related to a transgenerational transmission of this exposure to offspring born to parents who experienced food scarcity during WWII (1939-1945) during their gestation. Among the 1961-1966 cohort, their mothers were born between 1917-1950 (1st quartile: 1933, median: 1937, 3rd quartile: 1940). Moreover, among the mothers of the 1961-1966 cohort, 34% were born during WWII, while 13% were born between 1943-1948 (Fig. S6).”

Additionally, this is reiterated and emphasized in the limitations section as follows.

“Despite the small sample of the Dutch famine methylation study and limited evidence of transgenerational effects, we felt that the similarities between the 1943-48 and 1961-66 cohorts warranted exploring epigenetic mechanisms as one possible – albeit tentative – explanation. As expected, among the participants born in 1961-1966, two-thirds of their mothers were born outside the WWII period. Therefore, a potential transgenerational transmission from participants whose mothers were exposed to WWII alone cannot explain the findings in the 1961-1966 cohorts, which must be treated as tentative.”

Fig. S6. Distributions for mothers' year of birth, for the four cohorts.

REVIEWERS' COMMENTS

Reviewer #1 (Remarks to the Author):

No further comments.

Reviewer #2 (Remarks to the Author):

I thank the authors for tempering their interpretation of results. I think the authors have addressed my comments.

I would suggest you may want to consider conducting the sensitivity analyses and sharing these with the community as a letter to the editor or a comment on the manuscript when the UKB eventually delivers the required data.

Reviewer #2 (Remarks on code availability):

Code is provided to map the expression data on build hg19 and set up the data for the calculation of the famine PGS.

The code uses standard R packages. I have not run the code.